# Evaluating the Impact of Text De-Identification on Downstream NLP Tasks

**Cedric Lothritz**    **Bertrand Lebichot**    **Kevin Allix**
**Saad Ezzini**    **Tegawendé F. Bissyandé**    **Jacques Klein**
University of Luxembourg
6, rue Coudenhove-Kalergi
L-1359 Luxembourg

{cedric.lothritz,bertrand.lebichot,kevin.allix
saad.ezzini,tegawende.bissyande,jacques.klein} @uni.lu

**Andrey Boytsov**    **Clément Lefebvre**    **Anne Goujon**
BGL BNP Paribas
10, rue Edward Steichen
L-2540 Luxembourg

{andrey.boytsov,clement.c.lefebvre,anne.goujon } @bgl.lu

## Abstract

Data anonymisation is often required to comply with regulations when transfering information across departments or entities. However, the risk is that this procedure can distort the data and jeopardise the models built on it. Intuitively, the process of training an NLP model on anonymised data may lower the performance of the resulting model when compared to a model trained on non-anonymised data. In this paper, we investigate the impact of de-identification on the performance of nine downstream NLP tasks. We focus on the de-identification and pseudonymisation of personal names and compare six different anonymisation strategies for two state-of-the-art pre-trained models. Based on these experiments, we formulate recommendations on how the de-identification should be performed to guarantee accurate NLP models. Our results reveal that de-identification does have a negative impact on the performance of NLP models, but it is relatively low. We also find that using pseudonymisation techniques involving random names leads to better performance across most tasks.

## 1 Introduction

Protection of personal data has been a hot topic for decades (Bélanger and Crossler, 2011). Careless sharing of data between companies, cyber-attacks, and other data breaches can lead to catastrophic leaks of confidential data, potentially resulting in the invasion of people's privacy and identity theft.

To mitigate damages and hold bad actors accountable, many countries introduced various laws that aim to protect confidential data, such as the Health Insurance Portability and Accountability Act (HIPAA) for healthcare confidentiality (Act, 1996), and the Gramm–Leach–Bliley Act (GLBA) in the financial domain (Cuaresma, 2002). Most notably, with the introduction of the General Data Protection Regulation (GDPR), the protection of personally identifiable information was codified into EU law. (Regulation, 2016) Failure to comply with these regulations can lead to huge fines in case of a data breach. Indeed, the amount of fines for GDPR violations adds up to over 1.5 trillion euros with the largest single fine of 746 million euros being imposed on Amazon.[1]

In order to mitigate data leaks, organisations such as financial institutes and hospitals are required to anonymise or pseudonymise sensitive data before processing them further. Similarly, automated NLP models should ideally be trained using anonymised data as resulting models could potentially violate a number of GDPR guidelines such as the individuals' right to be forgotten, and the right to explanation. Furthermore, models can be manipulated to partially recreate the training data (Song et al., 2017), which can result in disastrous data breaches. On the other hand, however, anonymisation of texts can lead to loss of information and meaning, making NLP models trained on anonymised data less reliable as a result (Meystre et al., 2014). Intuitively, this in turn could lead to a decrease in performance of such models when compared to models trained on non-anonymised

---

[1]at the time of writing this paper, according to https://www.privacyaffairs.com/gdpr-fines/

text. As such, it is crucial to choose an appropriate anonymisation strategy to lower this loss of information and avoid performance drops of models.

In this study, we investigate the impact of text de-identification on the performance of downstream NLP tasks, focusing on the anonymisation and pseudonymisation of person names only. This allows us to select from a wide array of NLP tasks as most datasets contain a large number of person names, whereas other types of names are less commonly found. Specifically, we compare six different anonymisation strategies, and two Transformer-based pre-trained model architectures in our experiments: the popular BERT (Devlin et al., 2018) architecture and the state-of-the-art ERNIE (Sun et al., 2020) architecture. Further, we look into nine different NLP tasks of varying degrees of difficulty.

We address the following research questions:

- RQ1: Which anonymisation strategy is the most appropriate for downstream NLP tasks?
- RQ2: Should a model be trained on original or de-identified data?

## 2 Experimental Setup

In this section, we present the datasets used in this study and we introduce the different anonymisation strategies that we compare against each other. We also show the pre-trained models we use.

### 2.1 Datasets

For this study, we selected several downstream tasks that greatly vary in complexity, ranging from simple text classification to complicated Natural Language Understanding (NLU) tasks featured in the GLUE benchmark collection (Wang et al., 2018). We ensured that each set contains a considerable number of person names. Most of these datasets are publicly available, except for a proprietary email classification dataset provided by our partners. Table 1 contains statistics about the datasets used for this study. We release the original as well as the de-identified datasets for most tasks.[2]

We choose three public classification tasks: Fake News Detection (FND)[3], News Bias Detection (NBD) (Bharadwaj et al., 2020), and Fraudulent Email Detection (FED) (Radev, 2008).

Five of our investigated tasks are featured in the GLUE collection, namely MRPC (Dolan and Brockett, 2005), RTE (Haim et al., 2006), WNLI (Levesque et al., 2012), CoLA (Warstadt et al., 2018), and MNLI (Williams et al., 2018).

Our final task is the Email Domain Classification Dataset (EDC) which we describe in greater detail. It is provided by our partners in the banking domain. As such, it is a proprietary dataset consisting of sensitive emails from clients, and thus cannot be publicly released. However, it serves as an authentic use-case for our study. The task consists of classifying emails along 19 broad domains related to banking activities such as *credit cards*, *wire transfers*, *account management* etc., which will then be forwarded to the appropriate department. We selected a subset of the provided dataset, such that each domain is represented equally. More specifically, for each domain in the set, we randomly selected $\simeq 500$ emails, for a total of nearly 9000 emails. Furthermore, the dataset is multilingual, but we perform our experiments on the emails written in French due to the high sample number.

### 2.2 Anonymisation Strategies

We consider six anonymisation strategies (AS1-6) for this study. These strategies are commonly found in the literature (Berg et al., 2020; Deleger et al., 2013). They largely fall into three categories: replacement by a generic token (AS1, AS2, AS3), removal of names (AS4), and replacement by a random name which we also refer to as pseudonymisation throughout this work (AS5, AS6). We describe each AS in Table 2. Table 3 shows the differences between each AS on an example.

### 2.3 Name Detection

In order to detect names in the datasets, we fine-tune a *BERT Large* model on the task of Person Name Detection. We use the CoNLL-2003 dataset for Named Entity Recognition (Sang and De Meulder, 2003) and modify it by relabeling every non-*Person* entity as non-entity. The resulting training set consists of 204 567 words, 11 128 are *Person* entities and 193 439 are labeled as non-entities.[4] The resulting model achieved an F1 score of 0.9694, precision of 0.9786, and a recall of 0.9694 on the modified CoNLL-2003 test set. We use this fine-

---

[2] https://github.com/lothritz/anonymisation_paper
[3] https://www.kaggle.com/shubh0799/fake-news

[4] The dataset used to to train the de-identification model can be found at https://github.com/lothritz/anonymisation_paper/tree/main/anonymisation_model

| dataset | FND | NBD | FED | MRPC | RTE | WNLI | CoLA | MNLI | EDC |
|---|---|---|---|---|---|---|---|---|---|
| train set | 4382 | 1374 | 8980 | 3668 | 2489 | 635 | 6039 | 39 999 | 6354 |
| dev set | 690 | 196 | 997 | 407 | 276 | 71 | 851 | 5000 | 926 |
| test set | 1237 | 395 | 1926 | 1725 | 800 | 146 | 1661 | 5396 | 1798 |
| #names | 68 890 | 15 610 | 30 404 | 3324 | 3685 | 898 | 2600 | 85 999 | 6550 |
| #unique | 7500 | 3247 | 6104 | 1729 | 2042 | 102 | 335 | 10 460 | 2807 |
| %de-identified | 90.9 | 83.9 | 55.7 | 43.1 | 51 | 61.9 | 41 | 93.8 | 42.6 |
| type | binary | multi | binary | binary | binary | binary | binary | multi | multi |

Table 1: Statistics for the datasets. Size of datasets, number of names found in the training set (#names), number of unique names found in the training set (#unique), percentage of samples that contains at least one name (i.e. the percentage of samples to be de-identified) (%de-identified), and the type of the classification task (binary/multiclass)

| Name | Description of AS |
|---|---|
| AS1 | Singular generic token |
| AS2 | Unique generic token for each name in document |
| AS3 | Unique generic token for each distinct name in document |
| AS4 | Removal of names |
| AS5 | Random name for each name in document |
| AS6 | Random name for each distinct name in document |

Table 2: Description of Anonymisation strategies

tuned model to detect and replace names from the training, validation, and test set of the selected downstream tasks.

## 2.4 Model Training

We compare the impact of de-identification strategies using two Transformer-based models: BERT (Devlin et al., 2018) and ERNIE (Sun et al., 2020). For the tasks written in English, we use the uncased BERT Base mode and the ERNIE Base models. For the EDC task, we use the multilingual mBERT model and the ERNIE-M model published by Ouyang et al. (2021). For our study, we use the Transformers library by Huggingface (Wolf et al., 2019) as our framework. Furthermore, we take a grid-search based approach to determine the most appropriate fine-tuning parameters for each downstream task (cf. Appendix A)

## 3 Experimental Results

In this section, we show the results of our experiments and address the research questions from Section 1. For each task and for each pre-trained model, we fine-tune a model on the original dataset and each of our six anonymised datasets. We also de-identify the test sets accordingly and evaluate each model on the corresponding test set. We do five runs for each case, and average the results. We then compare the average performance for each AS

to the performance of the models trained on original data. Table 4 shows the average performance of every model. For each of the GLUE tasks, we use the metric recommended by (Wang et al., 2018) and F1 score for the classification tasks.

### 3.1 Which anonymisation strategy is the most appropriate for downstream NLP tasks?

In order to determine the most appropriate strategy, we consider two ranking-based approaches: Borda Count and Instant Runoff (Taylor and Pacelli, 2008). For both approaches, we determine the score $s_{a,t}$ for each anonymisation strategy (AS, indexed by $a$) and for each task (indexed by $t$) in the following way: The best approach gets a score of five, the second best gets a score of four, etc.

The final *Borda Count* score for a given anonymisation strategy $A$ is defined as $\sum_{t=0}^{T} s_{A,t}$ (where $T$ is the total number of tasks, here, nine). The model with the highest total score is considered the best.

*Instant Runoff* is an iterative procedure. For each iteration, we count the number of wins for each AS, where an AS is considered a winner in a given task if its corresponding fine-tuned model outperforms every other model. We then eliminate the AS with the lowest number of wins and update the scores accordingly. We repeat this process until one AS remains, or until we cannot eliminate further ASs.

Table 5 shows the scores for each model and the winning anonymisation strategies according to the aforementioned approaches. For BERT models, we see that AS1, AS4, and AS6 are the best performing strategies according to Borda count, AS6 being a close winner. Instant Runoff leads to similar results with AS4 and AS6 reaching the final iteration, and AS6 being the overall winner. Furthermore, we note a lower variance in the scores for AS6

| | | |
|---|---|---|
| Original | "Hi, this is Paul, am I speaking to John?" | "Sorry, no, this is George. John is not here today." |
| AS1 | "Hi, this is ENTNAME, am I speaking to ENTNAME?" | "Sorry, no, this is ENTNAME. ENTNAME is not here today." |
| AS2 | "Hi, this is ENTNAME1, am I speaking to ENTNAME2?" | "Sorry, no, this is ENTNAME1. ENTNAME2 is not here today." |
| AS3 | "Hi, this is ENTNAME1, am I speaking to ENTNAME2?" | "Sorry, no, this is ENTNAME3. ENTNAME2 is not here today." |
| AS4 | "Hi, this is , am I speaking to " | "Sorry, no, this is . is not here today." |
| AS5 | "Hi, this is Bert, am I speaking to Ernie?" | "Sorry, no, this is Elmo. Kermit is not here today." |
| AS6 | "Hi, this is Jessie, am I speaking to James?" | "Sorry, no, this is Meowth. James is not here today." |

Table 3: Example for each anonymisation strategy

| | | BERT | | | | | | | ERNIE | | | | | |
|---|---|---|---|---|---|---|---|---|---|---|---|---|---|---|
| Task | Metric | Original | AS1 | AS2 | AS3 | AS4 | AS5 | AS6 | Original | AS1 | AS2 | AS3 | AS4 | AS5 | AS6 |
| FND | F1 | 0.973 | 0.976↑ | 0.974↑ | 0.969↓ | 0.965↓ | 0.968↓ | 0.971↓ | 0.968 | 0.962↓ | 0.960↓ | 0.960↓ | 0.956↓ | 0.956↓ | 0.963↓ |
| NBD | F1 | 0.653 | 0.658↑ | 0.647↓ | 0.654↑ | 0.681↑ | 0.674↑ | 0.683↑ | 0.678 | 0.681↑ | 0.684↑ | 0.695↑ | 0.709↑ | 0.653↓ | 0.669↓ |
| FED | F1 | 0.994 | 0.995↑ | 0.996↑ | 0.996↑ | 0.996↑ | 0.994 | 0.995↑ | 0.996 | 0.994↓ | 0.993↓ | 0.994↓ | 0.993↓ | 0.995↓ | 0.993↓ |
| MRPC | F1 | 0.791 | 0.786↓ | 0.769↓ | 0.768↓ | 0.797↑ | 0.792↑ | 0.783↓ | 0.811 | 0.824↑ | 0.817↑ | 0.799↓ | 0.832↑ | 0.826↑ | 0.820↑ |
| RTE | Acc | 0.691 | 0.670↓ | 0.654↓ | 0.639↓ | 0.624↓ | 0.644↓ | 0.666↓ | 0.703 | 0.696↓ | 0.665↓ | 0.671↓ | 0.683↓ | 0.716↑ | 0.676↓ |
| WNLI | F1 | 0.520 | 0.530↑ | 0.526↑ | 0.551↑ | 0.586↑ | 0.541↑ | 0.535↑ | 0.561 | 0.472↓ | 0.557↓ | 0.564↑ | 0.595↑ | 0.614↑ | 0.550↓ |
| CoLA | MCC | 0.555 | 0.520↓ | 0.522↓ | 0.524↓ | 0.443↓ | 0.495↓ | 0.532↓ | 0.519 | 0.517↓ | 0.543↑ | 0.556↑ | 0.385↓ | 0.540↑ | 0.542↑ |
| MNLI | Acc | 0.754 | 0.742↓ | 0.730↓ | 0.734↓ | 0.745↓ | 0.742↓ | 0.747↓ | 0.789 | 0.774↓ | 0.750↓ | 0.759↓ | 0.770↓ | 0.776↓ | 0.773↓ |
| EDC | F1 | 0.626 | 0.624↓ | 0.683↑ | 0.617↓ | 0.619↓ | 0.616↓ | 0.595↓ | 0.642 | 0.635↓ | 0.696↑ | 0.642 | 0.635↓ | 0.627↓ | 0.621↓ |

Table 4: Results of our fine-tuned models. We highlight in green (↑) the models that outperform the models trained on original data, in red (↓) the models that do not.

| | BERT | | | | | | ERNIE | | | | | |
|---|---|---|---|---|---|---|---|---|---|---|---|---|
| Task | AS1 | AS2 | AS3 | AS4 | AS5 | **AS6** | AS1 | AS2 | AS3 | AS4 | **AS5** | AS6 |
| FND | 5 | 4 | 2 | 0 | 1 | 3 | 4 | 3 | 3 | 1 | 1 | 5 |
| NBD | 2 | 0 | 1 | 4 | 3 | 5 | 2 | 3 | 4 | 5 | 0 | 1 |
| FED | 2 | 5 | 5 | 5 | 0 | 2 | 4 | 2 | 4 | 2 | 5 | 2 |
| MRPC | 3 | 1 | 0 | 5 | 4 | 2 | 3 | 1 | 0 | 5 | 4 | 2 |
| RTE | 5 | 3 | 1 | 0 | 2 | 4 | 4 | 0 | 1 | 3 | 5 | 2 |
| WNLI | 1 | 0 | 4 | 5 | 3 | 2 | 0 | 2 | 3 | 4 | 5 | 1 |
| CoLA | 2 | 3 | 4 | 0 | 1 | 5 | 1 | 4 | 5 | 0 | 2 | 3 |
| MNLI | 3 | 0 | 1 | 4 | 3 | 5 | 4 | 0 | 1 | 2 | 5 | 3 |
| EDC | 4 | 5 | 2 | 3 | 1 | 0 | 3 | 5 | 4 | 3 | 1 | 0 |
| Total | 27 | 21 | 20 | 26 | 18 | **28** | 25 | 20 | 25 | 25 | **28** | 21 |
| Avg. | 3 | 2.33 | 2.22 | 2.89 | 2 | **3.11** | 2.78 | 2.22 | 2.78 | 2.78 | **3.11** | 2.33 |

Table 5: Ranking scores for fine-tuned models. **Bold text** shows the winner according to Borda Count, underlined text according to Instant Runoff.

| | | BERT | | | | | | | ERNIE | | | | | |
|---|---|---|---|---|---|---|---|---|---|---|---|---|---|---|
| Task | Metric | Original | AS1 | AS2 | AS3 | AS4 | AS5 | AS6 | Original | AS1 | AS2 | AS3 | AS4 | AS5 | AS6 |
| FND | F1 | 0.973 | 0.933↓ | 0.910↓ | 0.907↓ | 0.950↓ | 0.963↓ | 0.963↓ | 0.968 | 0.951↑ | 0.938↓ | 0.935↓ | 0.957↑ | 0.967↑ | 0.967↑ |
| NBD | F1 | 0.653 | 0.566↓ | 0.551↓ | 0.546↓ | 0.601↓ | 0.602↓ | 0.609↓ | 0.678 | 0.683 | 0.684 | 0.659↓ | 0.687↓ | 0.683↑ | 0.683↑ |
| FED | F1 | 0.994 | 0.995 | 0.995 | 0.995 | 0.996 | 0.996 | 0.996 | 0.996 | 0.995 | 0.995 | 0.995 | 0.996 | 0.996 | 0.996 |
| MRPC | F1 | 0.791 | 0.809↑ | 0.811↑ | 0.811↑ | 0.819↑ | 0.816↑ | 0.814↑ | 0.811 | 0.848↑ | 0.848↑ | 0.849↑ | 0.852↑ | 0.804↓ | 0.834↑ |
| RTE | Acc | 0.691 | 0.665↓ | 0.663↑ | 0.669↑ | 0.670↑ | 0.645↓ | 0.660↓ | 0.700 | 0.703↑ | 0.701↑ | 0.693↓ | 0.699↑ | 0.688↓ | 0.704↑ |
| WNLI | F1 | 0.520 | 0.504↓ | 0.504↓ | 0.504↓ | 0.504↓ | 0.504↓ | 0.504↓ | 0.561 | 0.435↓ | 0.442↓ | 0.467↓ | 0.506↓ | 0.458↓ | 0.428↓ |
| CoLA | MCC | 0.555 | 0.376↓ | 0.515↓ | 0.528↓ | 0.335↓ | 0.549↓ | 0.550↑ | 0.519 | 0.427↓ | 0.537↑ | 0.511↓ | 0.313↓ | 0.518↓ | 0.523↓ |
| MNLI | Acc | 0.754 | 0.753↑ | 0.724↓ | 0.753↑ | 0.753↑ | 0.744↓ | 0.744↓ | 0.789 | 0.783↑ | 0.545↓ | 0.760↑ | 0.772↑ | 0.669↓ | 0.765↓ |

Table 6: Results of testing the original models on de-identified data. We highlight in green (↑) the models that significantly outperform the matching model in Table 4 using a Wilcoxon test, in red (↓) the models that perform significantly worse, in black the models that do not perform significantly differently.

when compared to AS4. In contrast, when evaluating ERNIE models, we note that AS5 models are performing significantly better than every other strategy according to Borda Count. Similarly, AS5 also wins the Instant Runoff with AS4 and AS5 making it to the final round. Overall, it appears that using random names over generic tokens to de-identify textual data is the preferable solution as AS1, AS2, AS3 models, which were all trained

on data with generic tokens, usually rank low.

## 3.2 Should a model be trained on original or de-identified data?

In order to answer this question, we investigate the performance of models trained on original data on the de-identified test sets (cf. Table 4) and compare them to the models trained directly on de-identified data. Table 6 shows the results of testing models

trained on original training sets and evaluated on each of the de-identified test sets. We find that nearly half of the models trained on de-identified data outperform the counterpart model trained on original data. While there is not always a clear trend, we observe that the original models almost consistently perform better in the MRPC and RTE tasks, and perform worse in the WNLI and CoLA tasks, regardless of the architecture used. Furthermore, for BERT models, the models trained on de-identified data consistently perform worse on the FND and NBD tasks. For the ERNIE models, the models trained on original data consistently perform better on the FED task ever so slightly. Despite these observations, we notice that the performance losses are oftentimes very high, specifically for the NBD, WNLI, and CoLA tasks, while performance gains tend to be lower.

## 4 Discussion

Judging by the results of our experiments, we recommend practitioners to de-identify their sensitive textual data using random names, as they typically lead to the best results among the anonymisation strategies we tested. We also recommend to de-identify data before the training of NLP models. It follows that it is important to keep the de-identification process and naming schemes consistent throughout the entire pipeline that uses the data in order to mitigate potential performance losses of models. It may also be important to keep the number of names sufficiently high in order to avoid introducing bias in the training that may contribute to unfair discrimination against specific names, a well-known issue in machine learning models that handle person names (Caliskan et al., 2017).

## 5 Related Work

Relevant studies done on textual data largely focus on medical texts and on a very limited number of tasks and anonymisation strategies when compared to our work. On the other hand, they typically anonymise a wide variety of protected health information (PHI) classes, while our work focuses on anonymisation of persons' names only. Berg et al. (2020) studied the impact of four anonymisation strategies (pseudonymisation, replacement by PHI class, masking, and removal) on downstream NER tasks for the clinical domain. Similarly to our findings, they find that pseudonymisation yields the best results among the investigated strategies.

On the other hand, removal of names resulted in the highest negative impact on the downstream tasks. Deleger et al. (2013) investigated the impact of anonymisation on an information extraction task using a dataset of 3503 clinical notes. They anonymised 12 types of PHI such as patients' name, age, etc., and used two anonymisation strategies (replacement by fake PHI, and masking). They found no significant loss in performance for this task. Similarly, Meystre et al. (2014) found that the informativeness of medical notes only marginally decreased after anonymisation, using 18 types of PHI and 3 anonymisation strategies (replacement by fake PHI, replacement by PHI class, and replacement by *PHI* token). Using the same anonymisation strategies and ten types of PHI, Obeid et al. (2019) investigated the impact of anonymisation on a mental status classification task. Comparing nine different machine learning models, they did not find any significant difference in performance between original and anonymised data.

## 6 Conclusion

In this paper, we conducted an empirical study analysing the impact of de-identification on downstream NLP tasks. We investigated the difference in performance of six anonymisation strategies on nine NLP tasks ranging from simple classification tasks to hard NLU tasks. Further, we compared two architectures, BERT and ERNIE. Overall, we found that de-identifying data before training an NLP model does have a negative impact on its performance. However, this impact is relatively low. We determined that pseudonymisation techniques involving random names lead to higher performances across most tasks. Specifically, replacing names by random names (AS5) had the least negative impact when using an ERNIE model. Similarly, replacing by random names while preserving the link between identical names (AS6) worked best for BERT models. We also showed that it is advisable to de-identify data prior to training as we observed a large difference in performance between models trained on original data versus de-identified data. There is also a noticeable difference between the performances of BERT and ERNIE, warranting further investigation into the performance differences between a larger number of language models.

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

# 7 Appendices

## 7.1 Appendix A: Fine-Tuning Hyperparameters

| Task | BERT | | | ERNIE | | |
|------|------------|------------|--------|------------|------------|--------|
|      | batch size | learn rate | #epoch | batch size | learn rate | #epoch |
| FND  | 16 | 5e-5 | 1 | 8   | $2^{-5}$ | 1 |
| NBD  | 16 | 5e-5 | 3 | 8   | $2^{-5}$ | 5 |
| FED  | 32 | 3e-5 | 3 | 32  | $5^{-5}$ | 1 |
| MRPC | 16 | 5e-5 | 3 | 32  | $3^{-5}$ | 4 |
| RTE  | 16 | 5e-5 | 4 | 4   | $2^{-5}$ | 4 |
| WNLI | 16 | 3e-5 | 4 | 8   | $2^{-5}$ | 4 |
| ColA | 16 | 5e-5 | 3 | 64  | $3^{-5}$ | 3 |
| MNLI | 16 | 5e-5 | 2 | 512 | $3^{-5}$ | 3 |
| EDC  | 16 | 5e-5 | 5 | 8   | $3^{-5}$ | 3 |

Table 7: Hyperparameters for fine-tuning pretrained models for downstream tasks