# OpenReview forum: "Evaluating the Impact of Text De-Identification on Downstream NLP Tasks"
_NoDaLiDa/2023/Conference — NoDaLiDa 2023_

### Official Review · Reviewer_bHue · 2023-02-27
**An impressive volume of tests**

**Rating:** 6
**Confidence:** 4

**Review:**

This paper describes a systematic comparison of NLP task performance on anonymised data, while varying several different parameters: nine NLP tasks, two learning models, six anonymisation strategies, and with or without anonymising the training data.

The background is well written, explaining the things that need explaining and smoothly omitting the things that could not be well explained in a short paper. The one thing that raises eyebrows is that the five tasks from the GLUE set are only mentioned by name, without any further explanation. It would seem reasonable to mention with a word or two what the actual task is – for example "entailment (RTE)", much like for the non-glue tasks. As far as I can tell, the text also neglects to mention which languages are used (other than in the last case), and whether all tasks use different datasets.

The experiments seem to be thoroughly done, and the sheer number of tests is admirable. The results are not unexpected, but will probably be of significant practical use. This is enough reason the paper should be published.

The weaker part of the paper is the analysis and reasoning based on the results. Not much is said about the reliability of the results, and they do raise a few questions. Looking at table 3, we see that several of the results are actually better than without anonymisation, which is surprising, but we also see several cases where one model performs better and the other worse, so surely there is quite a lot of randomness involved here. Some idea of the significance of these differences would have been nice.

Table 5, showing the "Results of testing the original models on anonymised data" (should that be "training"?), briefly mentions a Wilcoxon test. That is good, but again we see several cases where the result is apparently significantly better with one model and significantly worse with the other, and there is overall no obvious pattern to the results, so one wonders if the effects really are reliable.

The discussion also has little to say about reliability, or about why the results are what they are. For example, anonymisation 6 consists of replacing names with other names, which apparently has real consequences: 4 of the 9 tasks give worse results for both models, 3 give better results for the first but worse for the second, and two give better results for the second but worse for the first. This looks to me like mostly random effects, but statistics aside, what would be the reason for the difference? Is the new set of names noticeably different from the first, and if so, could that not be changed?

We can also compare with AS3, which does essentially the same thing but in all caps. Are the models significantly worse at dealing with names if they are all spelled "ENTNAME" plus a number? That seems like a flaw in the models that could be fixed – identifying something as a name should not be more difficult if it is marked in a systematic way.

Minor comments:

The citation for GDPR should probably look different. It is also a little odd that the statement that it "was codified into EU law in 2018" is supported by a citation from 2016.

In table 3, a couple of the values are given with two significant digits instead of three.

**Paper Type:**

Short paper

---

### Official Review · Reviewer_5ctu · 2023-03-04
**Accept with a few revisions**

**Rating:** 8
**Confidence:** 4

**Review:**

The article is very timely, and experiments are illuminating, although not sufficiently explained. Given that we may expect a requirement that nearly all research data be subject to some kind of anonymization / pseudonymization before made open, it is important to know the effects of anonymization / pseudonymization both on the data itself, models trained on such data and data usefulness for further research.

Although experiments are performed on English and partly on French, the conclusions are generalizable to ither languages. However, focus on other languages would be interesting and I would like to encourage the authors to consider other languages, especially Scandinavian ones.

There are a few necessary improvements to the paper:

First of all, the experimental setups (for RQ1 and RQ2) need to be better explained. I assume that an extra page following the reviews can be allowed (and if not, could be argued for) to give the space. The results promise to be of importance to the community and the paper has a potential to be well-cited, provided it has better explanations.
Do I understand correctly that you take e.g. COLA dataset, anonymize it in six different ways so that you get seven versions of the dataset. Then, you train seven different BERT-based models and seven EARNIE-based models for the COLA-task, and test them – on what? On the original data (held-out testset)?

Second, some of the measures you use are not common knowledge (not for me, at least), such as Borda Count and Instant Runoff. You should explain better what they mean, and what their scores mean? I got confused by the fact that a zero (“0”) was both in bold and underlined as a winner, although in the text it is mentioned that “the best approach gest a score of five, ...” – so how a zero could be a winner? Again, the results reported in table 4 – what were the models tested on?

Third, In Section 3.2 you say that “Table 5 shows the results of testing models trained on non-anonymized training sets on anonymized  test sets”  all 6 versions of those (AS1-6)?

Fourth, please split your conclusions into the two RQs.

Finally, I would like to hear a bit of discussion with relation to the following questions:
- Which way do you propose to anonymize original data (i.e. in which direction  do your results point you to)?
- What losses in models can we expect if we use non-original data?
- Elaborate on whether you expect that any biases may be introduced into models if you apply randomized pseudonymization?

Good luck on your extremely important work!


**Paper Type:**

Short paper

---

### Official Review · Reviewer_sqmU · 2023-03-11
**narrow contribution with a few shortcomings**

**Rating:** 6
**Confidence:** 4

**Review:**

The paper provides an empirical analysis of how various strategies for replacing direct identifiers (in this case, person names) affect the performance of downstream tasks. The paper is clear and well-written, although the contribution is relatively narrow.

I do have a number of comments:
1) anonymization is not the right term to employ here, as the strategies describe fall far short of what would be necessary for GDPR-compliant anonymization. Indirect identifiers (quasi-identifiers) are ignored from the analysis, and even other type of direct identifiers beyond person names are ignored. And even if all of those identifiers were to be masked, this would in many case not account for a full GDPR-compliant anonymization (for a discussion, see Weitzenboeck, E. M., Lison, P., Cyndecka, M., & Langford, M. (2022). The GDPR and unstructured data: is anonymization possible?. International Data Privacy Law, 12(3), 184-206).
2) more appropriate terms would be text de-identification, text sanitization, or pseudonymization. The last term is employed several times in the article, but is never defined -- it would be useful to define it earlier on.
3) I couldn't find anywhere how the person names were actually detected in those texts. Did you run a NER model? Perform this task manually? Or were those already annotated in the datasets?
4) An important limitation of the present work is that it only considers text classification tasks. Those tasks can often be learned by detecting linguistic patterns that are typical of each category, for which the occurrence of person names, while useful, is unlikely to play a decisive role. I would assume that the results would have been different if the downstream tasks had focused on problems such as question answering or information extraction, where the task performance requires access to detailed semantic content.
5) Since one of the goals of the presented approach is to obtain more "privacy-aware" NLP models (with a lower risk of privacy leakages), the paper should at least mention approaches based on differentially-private training, such as DP-SGD and its variants (for some pointers, see e.g. Brown, H., Lee, K., Mireshghallah, F., Shokri, R., & Tramèr, F. (2022, June). What Does it Mean for a Language Model to Preserve Privacy?. In 2022 ACM Conference on Fairness, Accountability, and Transparency, pp. 2280-2292).

Last detail: personal names -> person names

**Paper Type:**

Short paper

---

### Decision · Program_Chairs · 2023-03-17

Accept